# The experience of being a caregiver of patients with digestive cancer, from patients and caregivers' perception: A mixed study

**Charlotte Grivel**[1], **Pierre Nizet**[1,2], **Manon Martin**[1], **Solange Pécout**[3], **Aurélie Lepeintre**[4], **Yann Touchefeu**[3], **Sonia Prot-Labarthe**[1,5], **Adrien Evin**[2,4], **Jean-François Huon**[1,2]*

1 Nantes Université, CHU Nantes, Pharmacie, Nantes, France, 2 Nantes Université, University Tours, CHU Nantes, CHU Tours, INSERM, MethodS in Patients-Centered Outcomes and HEalth Research, SPHERE, Nantes, France, 3 Institut des Maladies de l'Appareil Digestif, Nantes Université, CHU Nantes, Oncologie Digestive, Nantes, France, 4 Nantes Université, CHU Nantes, Service de soins Palliatifs et de Support, Nantes, France, 5 Université Paris Cité, Inserm, ECEVE, Paris, France

* jeanfrancois.huon@chu-nantes.fr

**Data Availability Statement:** Data for quantitative study are available as Supplemental material. For data collected as part of the qualitative research,

## Abstract

### Backgrounds

Caregivers are essential in the care of a patient with digestive cancer. Considering their experience and needs is crucial.

### Objectives

To explore the experience of caregivers of patients with digestive cancer and to compare the perspectives of patients and caregivers.

### Methods

A mixed-methods study with a cross-sectional prospective and a comprehensive qualitative dimension was performed in a medical oncology unit in a French tertiary hospital. Dyads made of patients with digestive cancer and their caregiver were recruited. The Caregiver Reaction Assessment (CRA) and the Supportive Care Needs Survey for Partners and Caregivers (SCNS-PC) questionnaires were distributed to caregivers. The CRA was used to measure the caregiver burden and the SCNS-PC was used to identify the unmet supportive care needs of caregivers. Semi-structured interviews with the dyads were conducted. Qualitative interviews addressed various dimensions of the caregiver's experience from each dyad's member perspective.

### Results

Thirty-two caregivers completed the questionnaires. Responses showed high self-esteem, schedule burden, and a need for care and information services. Ten dyads participated in the interviews. Three themes emerged from the caregiver's interviews: illness is an upheaval; loneliness and helplessness are experienced; caring is a natural role with positive outcomes. Four themes emerged from patient's interviews: the caregiver naturally assumes

we made excerpts of the transcripts relevant to the study available in the manuscript and in the supplemental data. Sharing the full transcripts would violate the agreement to which the participants consented.

**Funding:** The authors received no specific funding for this work.

**Competing interests:** The authors have declared that no competing interests exist.

the role and gets closer; he is the patient's anchor; his life is disrupted; anxiety and guilt accompany the desire to protect him. In comparing patient and caregiver data, the main theme of disagreement was their relationship.

## Conclusions

Caregiver care does not appear to be optimal, particularly in terms of their need for information. Patients have a fairly good representation of their experience, but the caregivers' opinion need to be considered.

## Introduction

Caregivers are critical to the care of cancer patients and often underprepared for this role [1]. Among other things, they provide support for activities of daily living, administrative tasks, as well as financial and emotional support, and they are frequently involved in symptoms management or treatment administration for their relatives [2,3]. Globally, caregivers are instrumental in caring for family members and loved ones, to the point that they experience burden when caring for them. There are many different interpretations of caregiver burden in the literature, but it can be defined as "*all the material and moral constraints that the dependence of a loved one places on the caregiver and their consequences on his or her physical and psychological health*" [4]. This translates physically and on their global health: more fatigue, sleep disorders, or else pain [2,5]. Psychologically, higher levels of anxiety and depression are experienced by caregivers [2,5,6]. In addition, their social and family life is directly impacted, sometimes leading to social isolation [2,3]. Their schedule, modified by that of their relative, no longer allows them to enjoy their leisure time [3,5]. Their professional and financial lives are not spared either [5,7,8]. Many end up forgetting their own needs [1,3,9]. Yet, meeting these needs is essential to continue properly caring for their loved one. Especially with the rise of outpatient treatment and the improvement of patient survival, caregivers assume this role for longer and longer. Despite that, caregivers can find positive aspects, like a sense of accomplishment or personal gratification, as well as the feeling of giving new meaning to their lives. In addition, some see their relationship with the patient improve [2,4,5,10].

Digestive cancers, like gastric cancer, pancreatic cancer, and cholangiocarcinoma, are among the most common cancers [11]. Their treatment results in varying levels of symptom distress, decline in social function, and disease-related anxiety, which may result in unmet supportive care needs [12] that may be sources of greater involvement for family caregivers. As seen before, experience of caregivers of cancer patient's is widely described in literature. However, asking for the patient's perspective on what their caregiver is experiencing has been less studied. The limited literature on this topic has shown that patients with head and neck cancer felt that their care was a considerable burden and that it was very hard for their caregiver [13]. To date, few studies have focused on caregivers of patients with digestive cancers but Mosher showed that caregivers and colorectal cancer patients failed to identify the same challenges that caregivers face [14–16].

This study aimed to explore the experience of being a caregiver of digestive cancer patients and compare the patient and caregiver perspectives to highlight differences and similarities. A mixed-methods approach with both qualitative and quantitative sides was used.

## Materials and methods

### Study design

A two-part mixed-study was carried out to provide a deeper understanding of caregivers' experiences [17]. The quantitative part was a cross-sectional single-center prospective study. The qualitative part was descriptive and comprehensive, using a phenomenological approach [18]. A convergent design where quantitative and qualitative data collection was simultaneous was used. The analysis of the two types of data was carried out separately. in the design, or conduct, or reporting, or dissemination plans of the research.

### Participants and recruitment

The study was conducted in a medical oncology unit and its day hospital in a French tertiary hospital between February and May 2022. Patient-caregiver dyads were recruited on the basis of patients' visits to consultations or day hospitals. Caregivers were approached either by mail, telephone, or face-to-face. The inclusion criteria for patients were to be aged 18 years and older and to be diagnosed with digestive cancer. The inclusion criteria for caregivers were to be aged 18 years and over and to be designated as the primary caregiver by a patient with digestive cancer. Non-inclusion criteria were refusal to participate by any member of the dyad or inability to complete a questionnaire or participate in an interview (cognitive impairment, language barrier, reading difficulties). For all participants, sociodemographic variables were collected (age, gender, type of relationship, socio-professional category). For patients, their clinical situation (location, metastatic status, performance status (PS)) was extracted from medical records.

### Quantitative data collection and analysis

The "Strengthening the Reporting of Observational Studies in Epidemiology (STROBE) tool [19] guided the reporting in this study. A cross-sectional study based on two validated questionnaires was conducted with caregivers. The Caregiver Reaction Assessment (CRA) was used to measure the caregiver burden [20]. It was used in its validated French version [21], which includes 24 items structured into five burden domains: caregiver self-esteem (seven items), schedule burden (five items), lack of family support (five items), health burden (four items), and financial burden (three items). Caregivers responded using a five-point Likert scale (1 = strongly disagree to 5 = strongly agree). For each burden domain, a score was calculated. The Supportive Care Needs Survey for Partners & Caregivers (SCNS—PC) was used to identify the unmet supportive care needs (USCN) of caregivers [22]. We used its validated French version [23], which includes 41 items classified into four types of needs: health care service and information (18 items), emotional and psychological (16 items), work and social security (four items), and communication and family support (three items). Caregivers rated the need for help on a five-point scale that differentiates between no needs (1 = not applicable, 2 = fulfilled needs) and USCN (3 = low, 4 = moderate, 5 = high). We used the analysis method of the French team that validated the questionnaire, recoding answers 1 and 2 as 1 (no need) and responses 3 to 5 as 2 to 4. For each type of need, a score was calculated. A higher score meant more need for help.

Data were anonymized and descriptive statistics were performed to summarize responses in terms of frequencies, percentages, means, standard deviations (SDs), ranges, and medians, as appropriate.

### Qualitative data collection and analysis

The Consolidated Criteria for Reporting Qualitative Research (COREQ) checklist [24] guided the reporting in this study. The patients and their caregivers participated in individual semi-

structured interviews. The first author conducted the interviews, in the hospital for the patients and by telephone primarily for the caregivers. She was trained to conduct an interview beforehand during a simulation session with the last author trained in qualitative research. The focus groups were conducted in French, and no non-participants was present during the interview. Two semi-structured interview guides constructed from the literature were used, one for caregivers and one for patients (S1 File). They consisted of open-ended questions allowing the interviewee to express himself freely on the topic of caregiving for patients with digestive cancers. They addressed the same themes: physical and psychological state of the caregiver; impact on daily life and difficulties encountered or positive experiences lived; impact on the caregiver-patient relationship; vision of the future. The interviews were recorded (audio) and then transcribed and analyzed independently by the first and third authors. The qualitative analysis was then carried out by the first and third, using the six-step thematic analysis by Braun and Clarke [25]. Briefly, it consisted of categorization and thematization of all the verbatims from each group through systematic inductive coding. The authors compared their analyses until they reached a common interpretation, following four stages of analysis (immersion in the data, coding, creation of categories, and identification of themes). NVIVO V.11 software was used for coding. Data collection was continued until saturation was met, i.e. when no new categories appeared in the thematic analysis [26]. After anonymization, each dyad was assigned a number, and a letter was used to differentiate caregivers (C) and patients (P). This allowed the creation of an identification code (ID) for each participant.

The interviews with patients and caregivers were analyzed separately to consider the perspective of each. Then, each dyad was studied and compared one by one, to detect possible differences or similarities within the dyad.

### Ethics statement

Informed consent was obtained from each patient and caregiver included. For the interviews, written consent was obtained. Ethical approval was granted by a local ethics committee (*Groupe Nantais d'Ethique dans le Domaine de la Santé)* on February 16, 2022 (#GNEDS20220216).

## Results

### Quantitative results

The questionnaires were distributed to 63 caregivers. Thirty-two were returned, representing a response rate of 50.8%. The characteristics of the participants are described in Table 1.

The mean age of the caregivers was 56.1 years (SD, 19.0), and 71.9% were female. Most (81.3%) lived with the patient and 71.9% were their spouse. 93.8% of them reported providing emotional support, and 78.1% helping with household tasks. The majority (71.0%) were not working, either retired or not employed. The patients they cared for were mainly male (62.5%) with a mean age of 63.3 years (SD, 11.9). Most patients (71.9%) had a good PS (0–1).

Based on CRA responses, caregivers reported a relatively high score in the self-esteem domain (mean, 3.8; SD, 0.7). The highest burden was the impact on schedule (mean, 3.3; SD, 1.0), followed equally by health (mean, 2.7; SD, 0.8) and financial impacts (mean, 2.7; SD, 0.9). Lack of family support ranked last in their concerns (mean, 2.4; SD, 0.7).

According to SCNS-PC's responses, only two (6.3%) caregivers had no USCN, meaning that 93.7% of caregivers had at least one unmet need. On average, caregivers noted that 46.3% of the needs presented were unmet (19 of 41 needs items). Considering moderate to high USCN, 56.3% reported at least 10. Across the four domains of needs, the highest scores were found for healthcare service and information needs (mean, 2.0; SD, 0.8) followed by emotional

**Table 1. Demographics of the caregivers and their patients (N = 32 dyads).**

| Caregiver (N = 32) | | | | Patient (N = 32) | | | |
|---|---|---|---|---|---|---|---|
| Age, y†, mean +/- SD‡ | | 56.1 +/- 19.0 | | Age, y†, mean +/- SD‡ | | 63.6 +/- 11.9 | |
| Time being a caregiver, m§, min-max (median) | | 2–48 (7.5) | | Time since diagnostic, m§, min-max (median) | | 5–139 (15.0) | |
| | | N | % | | | N | % |
| Sex | Female | 23 | 71.9 | Sex | Female | 12 | 37.5 |
| | Male | 9 | 28.1 | | Male | 20 | 62.5 |
| Socio professional category | Farmer | 0 | 0 | Socio professional category | Farmer | 0 | 0 |
| | Artisan, shopkeeper, CEO | 1 | 3.1 | | Artisan, shopkeeper, CEO | 0 | 0 |
| | Executive and intellectual profession | 1 | 3.1 | | Executive and intellectual profession | 2 | 6.3 |
| | Intermediate profession | 3 | 9.4 | | Intermediate profession | 1 | 3.1 |
| | Employee | 2 | 6.3 | | Employee | 0 | .0 |
| | Worker | 2 | 6.3 | | Worker | 2 | 6.3 |
| | Retired | 17 | 53.1 | | Retired | 18 | 56.3 |
| | Non-working | 6 | 18.8 | | Non-working | 9 | 28.1 |
| Type of support provided | Emotional support | 30 | 93.8 | Tumor localization | Oesophagus | 4 | 12.5 |
| | Financial support | 9 | 28.1 | | Stomach | 2 | 6.3 |
| | Administrative support | 18 | 56.3 | | Colon | 14 | 43.8 |
| | Household chores | 25 | 78.1 | | Rectum | 2 | 6.3 |
| | Medical care | 6 | 18.8 | | Liver | 3 | 9.4 |
| Living arrangement | Living with patient | 26 | 81.3 | | Pancreas | 7 | 21.9 |
| | Not living with patient | 6 | 18.8 | Metastatis | Yes | 16 | 50.0 |
| Type of relationship | Spouse/partner | 23 | 71.9 | | No | 16 | 50.0 |
| | Children | 5 | 15.6 | Place of recruitment | Day hospital | 21 | 65.6 |
| | Family-in-law | 1 | 3.1 | | Hospital ward | 11 | 34.4 |
| | Siblings | 2 | 6.3 | ECOG PS¶ | 0 to 1 | 23 | 71.9 |
| | Ex | 1 | 3.1 | | ≥ 2 | 9 | 28.1 |

Abbreviations:

† y, years;

‡ SD, standard deviation;

§ m, months;

¶ ECOG PS, Eastern Cooperative Oncology Group performance status.

ECOG PS ≤ 1 indicates normal activity; ≥2, from partial to complete bed rest.

and psychological needs (mean, 1.9; SD, 0.6). In third and fourth place were the needs related to family support and communication (mean, 1.6; SD, 0.9) and work and social security (mean, 1.5; SD, 0.6).

For each questionnaire, the responses to each item are presented in Figs 1 and 2.

## Qualitative results

Of the 32 patient-caregiver dyads for which we had quantitative results, 10 gave their consent to participate in the interviews. The main reasons for declining the interviews were lack of time or because it seemed too intimate. The average duration of an interview was 35 [15–65] min for caregivers and 30 [15–57] min for patients. The characteristics of the participants are presented in Table 2.

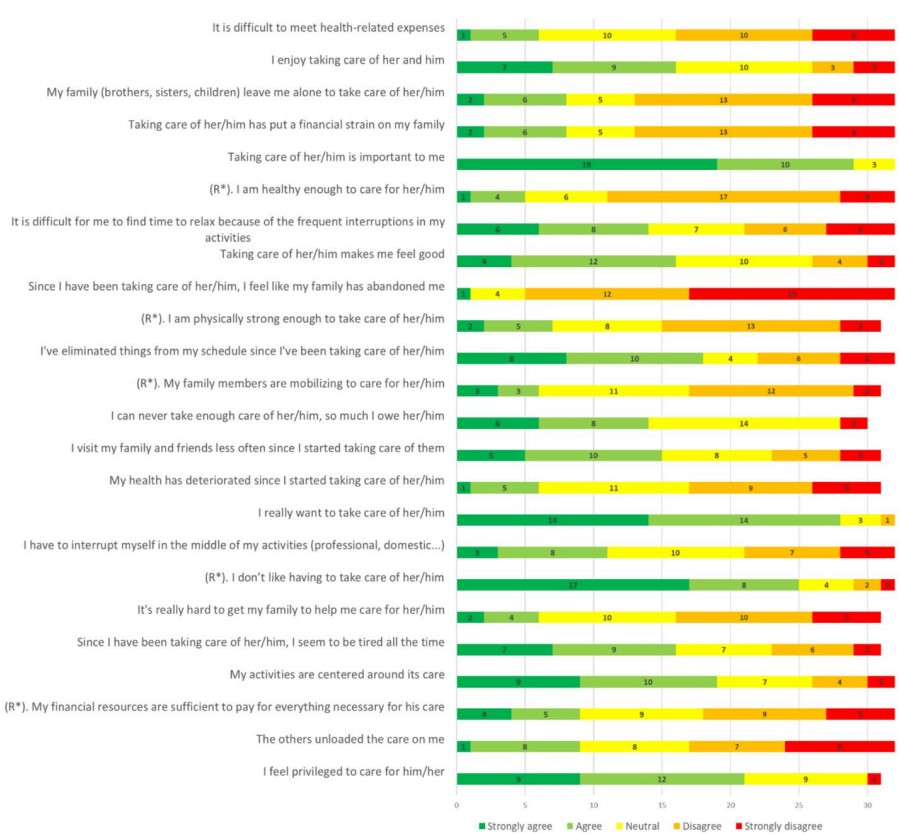

**Fig 1. Caregiver responses to CRA items (N: 30 to 32 responding to each item).** R*: reverse coded item.

The median age was 60.5 [19–86] years for caregivers and 67.5 [39–88] years for patients. Eight caregivers and four patients were female. Seven caregivers were spouses. One caregiver did not live with the patient. Two caregivers and one patient were employed. Four patients had a PS ≥ 2. The median duration of presence as a caregiver was 15 [2–39] months.

The themes and subthemes, along with illustrative quotes from the caregiver and patient interviews are presented in Tables 3 and 4 respectively.

Quotes are identified by the participant's anonymizing ID. Additional quotes are reported in S2 File.

**What is it like to be a caregiver? The caregiver's point of view.** *Illness is an upheaval in the caregiver's life.* Illness suddenly thrusts loved ones into the role of caregiver. This role disrupts their habits, changes their daily life (C25), and has an impact on their organization. This ranges from the distribution of household tasks to putting their studies on hold (C54) and being off work. They take on new roles within their own families by having to take care of their siblings (C54) or children (C15) on their own. Their habits are affected by financial problems or administrative procedures considered a waste of time. They are also forced to review their plans, they can no longer plan in the long term and must live in the present (C53). All of this is the consequence of the fact that the patient's illness is now their major concern (C33). All their free time are devoted to the patient so they change their schedule, put things aside (C01) and organize themselves around the illness. One of the things standing out is their desire to be involved and informed, they are curious (C12) about cancer and its evolution. All of this leads caregivers to forget themselves and their health: "My own needs [. . .] I don't forget them,

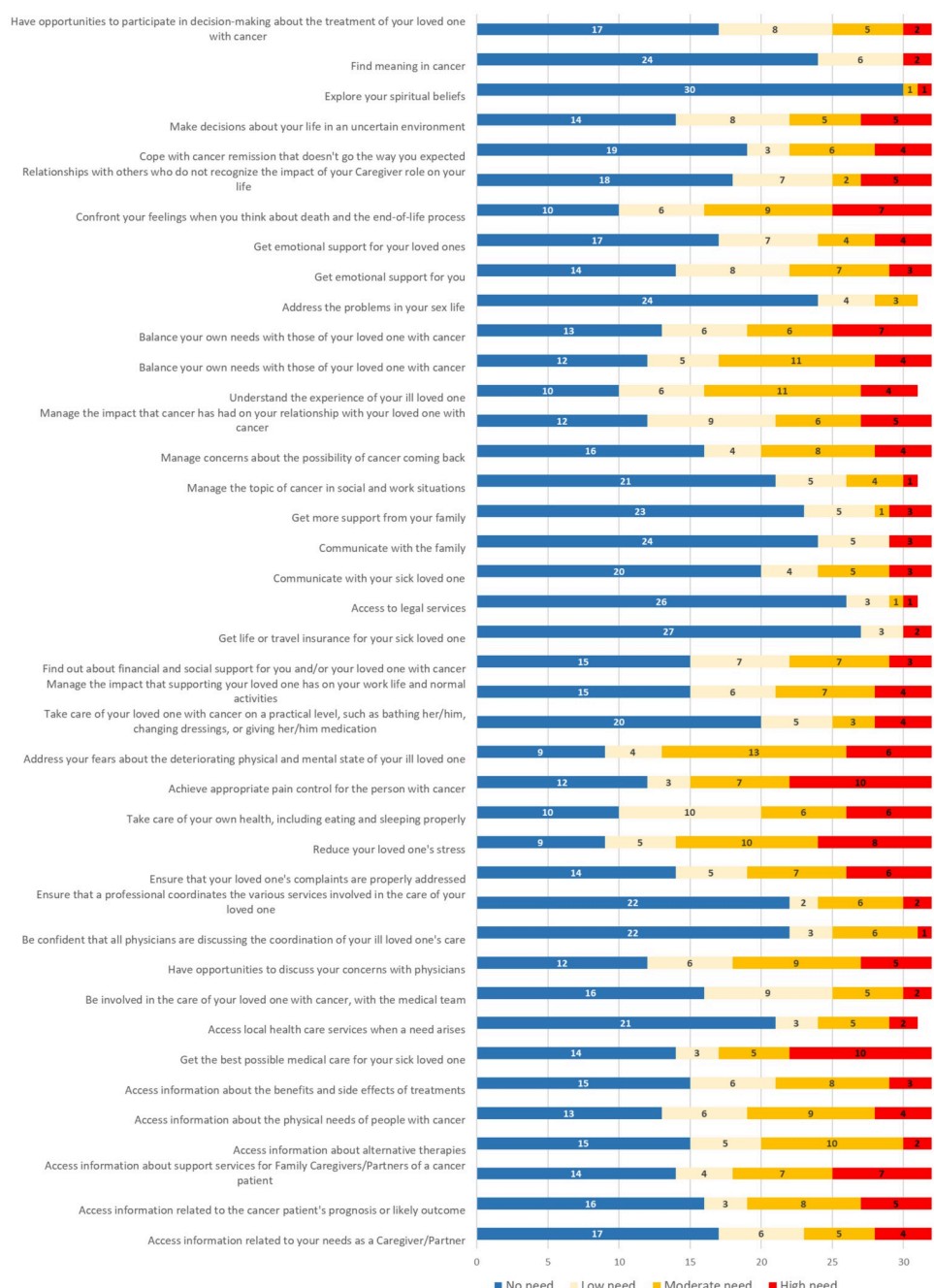

**Fig 2. Caregivers' responses to SCNS-PC items (N: 31 or 32 responding to each item).**

but I know how to put them in brackets for the time it takes" (C33). Finally, it weighs on them mentally and physically. They are drained (C06) and emotionally much sadder (C06). Because of the stress they often don't eat, they don't sleep (C15), so much so that they have to take pills, take naps during the day, or watch TV at night (C15, C26, C53). Sometimes they prefer to sleep alone in a room alone to sleep better (C53).

*He feels alone and helpless dealing with the disease and suffers its consequences.* The lack of control they have over the situation is difficult. Caregivers feel helpless in certain situations

**Table 2. General characteristics of the interview participant dyads (N = 10 dyads).**

| ID§ | Patient | | | | Relationship | ID§ | Caregiver | | | | |
|---|---|---|---|---|---|---|---|---|---|---|---|
| | Age, y† | Sex | Interview length *(min)* | Type of cancer | | | Age, y† | Sex | Interview length *(min)* | Socio-professional category | Time being a caregiver, m‡ |
| *P01* | 83 | M | 15 | Esophagus metastatic | Parent and child | *C01* | 59 | F | 15 | Executive and intellectual profession | 2 |
| *P06* | 76 | M | 23 | Colon non metastatic | Spouses | *C06* | 75 | F | 65 | retired | 14 |
| *P12* | 46 | M | 20 | Colon metastatic | Spouses | *C12* | 41 | F | 33 | Non-working | 16 |
| *P15* | 39 | F | 33 | Pancreas metastatic | Spouses | *C15* | 29 | M | 20 | Worker | 4 |
| *P25* | 66 | M | 37 | Colon metastatic | Spouses | *C25* | 62 | F | 35 | Retired | 22 |
| *P26* | 88 | M | 44 | Stomach non metastatic | Spouses | *C26* | 86 | F | 30 | Retired | 6 |
| *P33* | 52 | F | 57 | Colon non metastatic | Spouses | *C33* | 56 | M | 46 | Non-working | 36 |
| *P53* | 74 | M | 24 | Liver non metastatic | Spouses | *C53* | 74 | F | 53 | Retired | 39 |
| *P54* | 52 | F | 29 | Colon metastatic | Parent and child | *C54* | 19 | F | 25 | Non-working | 8 |
| *P51* | 73 | F | 20 | Pancreas non metastatic | Sisters | *C51* | 69 | F | 24 | Retired | 22 |

Abbreviations:

§ID, identification code;

ᵐ, months;

†y, years;

§ID, identification code.

and do not know how to act. The main source of helplessness is pain, "when she really suffers [. . .] but I can't do anything" (C51). Another difficulty is the unknown (C06) because they are often unprepared to deal with the progression of cancer, which can be fatal. At the same time, they also feel alone because they deplore the lack of time given to them by medical staff, and sometimes have to cry to get information from doctors (C06), they also raise the issue of a true caregiver status (C33). The illness can even affect their relationship with the patient because they do fewer activities together and tensions can arise between them (C06). At the level of intimacy, chemo also has an impact on their sexuality.

*Despite this, he naturally assumes his role while managing to remain positive.* Despite this, caregivers naturally assume their role. For them, it is normal (C25) and is not considered a constraint (C01). For spouses, it is a role inherent to their marital status (C12). It is also a way of giving back to the patient (C01). On the other hand, caregivers can draw positive things from the situation because it changes their outlook on life (C25), they see things more positively. It can make them feel useful (C33) and more accomplished (C12).

**What is it like to be a caregiver? The patient's point of view.** *The patient sees the caregiver take on his new role naturally and becoming closer to him.* From the patient's perspective, taking on this role is not something forced on the caregiver (P26), because they are married (P06) or because it is in the caregiver's values to put others first (P33). The caregiver also seems to be closer to them, their relationship would be strengthened (P12) and they would communicate more (P15).

*The caregiver becomes the anchor of the patient.* With the diagnosis of cancer, the patient needs more attention and over time sees himself becoming the center of attention of the

**Table 3. Themes and sub-themes from caregivers' interviews.**

| Theme/subtheme | Verbatims |
|---|---|
| **THEME 1: ILLNESS IS AN UPHEAVAL IN THE CAREGIVER'S LIFE** | |
| He must change his habits and re-plan the future. . . | "There are a lot of things that we used to share in the house, for example, he used to do crafts and there are a lot of things that he can't do anymore because he's too tired. So, I have to try to do it myself or I ask friends to help me. Obviously, it changes the daily life a little bit on that side. Before, we used to share the work, the house, the maintenance"—C25<br><br>"The organization was quite complicated. Between the children that I have to take to school and me with work. Of course, with the work schedule, it didn't match."—C15<br><br>"If I wasn't there. . . I can't imagine how the house would work if I wasn't there actually. . . and on top of that there's my 11-year-old brother" "I put my studies on hold, at least for this year."—C54<br><br>"We have to live. . .. We live in the present, while making plans that are cancelled as we go along."—C53<br><br>"My wife's illness is my overriding concern."—C33 |
| . . .because his life is now centered on the illness of his loved one. . . | "From time to time, I have to put things aside to be with him"—C01<br><br>"I know I didn't dare leave him too long alone. So I stopped all my activities"—C06<br><br>"I do what I can to make his life less, less difficult, at least I hope to make it less difficult. . . What he wants I do, I am very caring for him."—C26<br><br>"The consultations, I systematically accompany him, because. . . I am curious, well, curious to know how it goes, the diagnoses etc."—C12<br><br>"My own needs, how shall I say. . . I don't forget them, but I know how to put them in brackets for the time it takes"—C33<br><br>"I almost didn't have the operation because I thought. . . it was an operation that was planned, but as his condition had worsened in the meantime, I thought maybe I would cancel it."—C25 |
| . . .and it weighs on him mentally and physically. | " I take a lot on myself. So. . . I'm tired for sure, I feel drained." "I've gotten old, that's for sure. You see, I don't have the joy of living anymore. Normally, I'm always positive and quite. . . I'm always told that I'm very energetic and very cheerful and all that. Well, now when I am, it sounds fake."—C06<br><br>"In the beginning, I couldn't sleep. So I had pills to help me. Now everything is back, the sleep is back in order and much more intense. But I'm much more tired with all the things I have to do, and I have to be in bed by 8:30–9:00 p.m."—C15 |
| **THEME 2: HE FEELS ALONE AND HELPLESS DEALING WITH THE DISEASE AND SUFFERS ITS CONSEQUENCES** | |
| He has no control over the situation and does not know how to (re)act. | "Like for example, this week, she was re-hospitalized. And seeing her like this, it. . . I'm having a little trouble. . . I don't really know how to behave or react to that. . ."—C54<br><br>"When she really suffers, when the pain comes. . . I understand her when she's in pain, but I can't do anything about it, I can only maybe call the doctor or see if there is a painkiller. But as she's on a morphine patch, she has morphine, so it's just to check if every 3 days it's been changed or, you know, to help her in that sense."—C51 |
| He dreads what may happen. . . | "What is very difficult in these diseases is the unknown."—C06<br><br>"it's not always easy, when he's too bad, well, we think the worst. . . That's what's hardest. It's when you go to bed at night and you think. . . well, the nights when he was in hospital for three months and I came back at night. . . I went to see him almost every day, the children took me and we went almost every day. And when we came back in the evening, we said: "well I don't know, will we see him again tomorrow morning?""—C53 |

*(Continued)*

**Table 3.** (Continued)

| Theme/subtheme | Verbatims |
|---|---|
| . . .and does not feel heard. | *"I was never asked the question, "Do you need information, on this or that?". . . I even had to cry to get some information of the doctors."—C06*<br><br>*"It's not a recognized status. it seems to me, it's something a bit odd to be a caregiver nowadays, I have the impression that it goes a bit against the way our society is set up. We are rather in an individualistic society and the caregivers are a bit of aliens because they take care of other people. . . I think that all diseases need humanity so that people can recover. . . It would be nice to have a real caregiver status."—C33* |
| He sometimes sees his relationship with the patient deteriorate. | *"We find ourselves a bit alone, we find ourselves a bit alone between the two of us. Sometimes we get mad at each other. There is tension and we don't know how to do. . ."—C06*<br><br>*"Well, it hasn't really evolved. . . No, on the contrary. No. . . it was better before, in fact, before all that. . . because now we're both more on edge. . . both he and I. Because sometimes, and it's normal, he's angry to be sick, to not be able to do anything about it, to not be as independent as before. And sometimes it's me, I'm also more on edge, because I'm tired, more tired than before for sure. . . But here I am in the good and bad times. But our relationship. . . well it was better before"—C26* |
| **THEME 3**: Despite this, he naturally assumes his role while managing to remain positive | |
| He acts without asking too many questions. . . | *"It's something that's normal. I mean, I think it's normal in a couple that when one is not well, that the other one helps him. . . For me, it's normal, so I don't ask myself these questions. It's normal to help each other when one is sick. He is my husband, I love him, it is normal that I help him and he would have done the same for me."—C25*<br><br>*"it's not a constraint in the sense that I give back to my father what he gave us."—C01*<br><br>*"It's normal, it's natural. We are married, so it's common sense to be at your spouse's bedside. He would do the same if it were me. And that's it, for me, it's natural, it's common sense. I don't see this as a constraint, on the contrary"—C12* |
| . . .while managing to draw positive dimensions from it. | *"It changes the way we look at life, of course. At least that is beneficial, it makes us aware of how lucky we are for everything we had, for everything we have, and it makes us aware of all the times we complain about nothing. it reminds us that even if we're always complaining, we still have our health and that's what's important."—C25*<br><br>*"The positive is that we. . . Well, for example, a beautiful day of sunshine, a walk, that makes me, that makes me happy. You see? Things like that. Seeing my children at Christmas, I took advantage of it and I stored it up. Those moments when we were all together, I stored up. . . so in all the bad things you can take out the positive."—C06*<br><br>*"I feel useful, I help, I help my wife ""Well, being able to cheer her up or make her smile is. . . yes, these are rewarding moments."—C33*<br><br>*"It may not be the right word, "fulfillment" but it makes me feel good, in the sense that it is on the path of life you see?"—C12* |

caregiver. He sees that the caregiver is always worried about him (P25) and always wants to be with him: "She wants to go everywhere with me, if I don't forbid her to come, she comes. Even for one day of hospitalization, she comes." (P12). He sees how curious the caregiver is about his care because he asks many questions (P06) and sometimes more than the patient. The patient admits that he needs the caregiver, that he is indispensable to him (P26) because he does everything at home now (P15) and his presence in difficult moments is essential. In fact, the patient would not know what to do without the caregiver. They are grateful (P26) and even consider the caregiver as important as medical care (P33).

*But he notices that this disrupts the life of the caregiver*. The patient notes that all of this is putting a strain on the caregiver. He sees that the caregiver is not sleeping, is losing weight

**Table 4. Themes and sub-themes from patients' interviews.**

| Theme/subtheme | Verbatims |
|---|---|
| THEME 1: THE PATIENT SEES THE CAREGIVER TAKE ON HIS NEW ROLE NATURALLY AND BECOMING CLOSER TO HIM | |
| He sees the caregiver helping without asking too many questions. . . | *"She is a natural caregiver. . . it's not something that she forces herself to do. It's something, it seems to me, that's natural to her. Even if she doesn't express it in that way."—P26*<br>*"We are married, so she assumes her role as a married woman." "It seems to be quite natural for her, she is always willing to help others, including me."—P06*<br>*"He feels that this is his role." "He has always put the family before, in his values he has always put the family before. . ."—P33* |
| . . .and being closer | *"We talk a lot more. Yeah, we communicate a lot more. We try to communicate a lot more. So that we don't each stay in our own corner, so that we don't each close in on ourselves. And above all, so that we can both move forward." "It's true that I have the impression that we are stronger. . . Well, it makes us more. . . more solid than before in fact."—P15*<br>*"Our relationship, I think, has strengthened. We're more in tune with each other. We share more things."—P12* |
| THEME 2: THE CAREGIVER BECOMES THE ANCHOR OF THE PATIENT | |
| He sees himself becoming the center of attention of the caregiver. | *"She is always worried about me" "I know she's worried and she needs to be told things. Here, she called the doctor to find out exactly what the situation was, because nurses are not allowed to give details. And the doctor, who was very nice, took the time to explain it to her and that calmed her down. And for example, when I have check-ups with the oncologist, she asks to be present, to ask questions."—P25*<br>*"She wants to go everywhere with me, if I don't forbid her to come, she comes. Even for one day of hospitalization, she comes."—P12*<br>*"When I see the doctor and she's there, she asks a lot of questions, that way she's informed"—P06* |
| The patient recognizes needing him at all times. . . | *"I need her, I tell her. She is indispensable to me"—P26*<br>*"She accompanies me in the difficult moments or in the big moments of tiredness"—P12*<br>*"He does everything he can at home to. . . to keep everything running. He cleans, he does the dishes, he does the laundry, he does the kids, the school. He takes care of all that."—P15* |
| . . .s;o much so that doesn't know how he would do without him | *"I tell her that if she wasn't there, I probably wouldn't be there. She has extended my life, unquestionably, during these five-six years." "I'll always be grateful to her"—P26*<br>*"He is as important, I was going to say, as the medical. I don't want to disadvantage the medical, but it's true that for people who are not hospitalized 24 hours a day, caregivers are very important."—P33* |
| THEME 3: BUT HE NOTICES THAT THIS DISRUPTS THE LIFE OF THE CAREGIVER | |
| He witnesses the burden that this attention generates for the caregiver. | *"She has lost some weight" "the nights I have something, she doesn't sleep."—P53*<br>*"During the first 15 days he didn't sleep. So he had, he needed pills to sleep. Because during the day, he was taking care of the kids, the house and everything. And then at night, he spent his nights thinking."—P15*<br>*"She is very worried at the moment, because she has anxieties, she sleeps badly and the fact that I fell ill, it's serious, I have a rather serious cancer. It doesn't reassure her. It makes her anxious." "She is not very cheerful, it makes her sad, yes."—P06*<br>*"The previous treatment, I had clots in my implantable chamber and so he had to give me the injections for the anticoagulants. . . he gave me the injections and that was difficult for him. I know it's difficult, I can't ask him to do my care, it impacts him too much."—P33* |

*(Continued)*

**Table 4.** (Continued)

| Theme/subtheme | Verbatims |
|---|---|
| He sees that the caregiver is trying to keep up appearances with him. | *"I'm not fooled, I'm not fooled, no, no, he is not right. He's distraught"* *"He won't tell me that it affects him, even if he talks easily but that no, no, no... probably to protect me"—P33* *"He doesn't want to show anything, he keeps, he keeps everything to himself."—P15* *"She doesn't express her difficulties. I think she's doing everything she can... everything she can so that it doesn't show, towards me I mean... Presumably it is a desire to protect me"—P25* |
| **THEME 4: AND THIS WORRIES HIM, HE FEELS GUILTY TOWARDS THE CAREGIVER AND WOULD LIKE TO PROTECT HIM** | |
| He is concerned about the caregiver... | *"He forgets himself, so sometimes I tell him that there is also his life, there is also his daily life and that he must not forget himself in all that."—P33* *"The only time she really cried was when I was really bad. I was in front of our house, I couldn't even carry my bag, I rang the doorbell, I said "Come". When I got there, I threw up... I finally broke down, I was so unwell. And that's when she put herself in my arms and she cried, a lot. I had never seen her cry like that. Just thinking about it makes me feel like... It hurt my heart so much... I don't want to see her like that anymore."—P54* |
| ...and he feels guilty | *"Yes, there was a time when I felt like a burden."—P15* *"I would like not to be too heavy for her"—P25* *"For him it would be good if I wasn't there, because what am I going to do if I'm there and I'm not well? Well, I'm going to ruin that moment, it's ridiculous."—P33* |
| He would prefer that the caregiver accept to be helped... | *"He has to admit it too, that he needs help... it would free up his time... it would allow him not to have to think about everything all the time. But he has to get it into his head"—P15* *"I regret that she does not accept, for example, why not a help for... I say anything, a help for the meals, even if it is not simple what it is necessary to do for me. But there is still her, to make her eat and well she could... why not take someone two-three times a week to prepare the meals and always have some on hand."—P26* |
| ...so he tries to protect him in his way | *"I don't break down in their presence, the time I broke down, but because my body was failing me, I saw my daughter crying like crazy. So now when I cry, I cry when I'm alone."—P54* *"I try to make sure she has as little to do as possible. I try to assist her, at home, even if I can't do much of anything."—P06* *"Before, there were things that I hid from her, the big fatigue that I have during the treatment or the different side effects that I can have here (in the hospital)."—P12* |

(P53). Emotionally, he sees that it makes the caregiver sad (P06). When the caregiver has to provide medical care to the patient, he knows how much it can cost the caregiver because he is afraid of hurting the patient (P33). In addition, he realizes that it also affects their work life. The caregiver does not express the difficulties he is experiencing but the patient is not fooled (P33), he sees the caregiver hiding them from him, whether it is his need for help or his emotions. The patient suspects a desire to protect him and not to worry him (P25).

*And this worries him, he feels guilty about the caregiver and would like to protect him.* All this added up, the patient ends up worrying. He fears leaving the caregiver alone in case of a fatal outcome and laments that the caregiver sometimes forgets himself (P33). The patient then feels guilty and has the impression of being a burden (P15). Thus, he would like to lighten this burden and make the caregiver admit that he needs help (P15). The patient, therefore, tries to protect the caregiver in his way (P12) by hiding things about the disease (P12) or by making sure that the caregiver has as little to do as possible (P06).

**The differences and similarities of point of view between a patient and his caregiver.**
The results of the comparison of each dyad individually, along with illustrative quotes are presented in Table 5.

Quotes are only identified by a letter and not with participants' ID, to ensure no possible re-identification, given some of the topics discussed.

We consider similarities in the topics addressed in the same way by caregiver and patient regarding the experience of being a caregiver and differences when they express two real points of view that oppose each other. Thus, topics that are not common are not necessarily differences, some topics may not have been addressed by one of the dyad members.

For two dyads, we did not detect discrepancies. The main theme of the discrepancies detected for the other dyads concerned their relationship. For example, one patient felt that their relationship had improved, while the caregiver felt that it had grown apart. Another caregiver felt that the patient was not telling him everything, while the caregiver said that the patient was not hiding anything.

There were many themes on which the patient and caregiver of the same pair agreed, showing the patient's ability to correctly estimate the caregiver's experience, notably by considering the caregiver's burden.

## Discussion

This mixed-method study gave a general picture of the experiences of caregivers of digestive cancer patients. It shows the complex impact of caring for a patient with digestive cancer on the caregiver, and the difference in vision that a patient can have on these consequences compared to his caregiver.

Disruption of schedule was noted as the greatest burden from CRA's responses (score = 3.3). The same trend was found for colorectal cancer caregivers in a study also using this questionnaire (score = 3.05) [27]. Other studies have shown the importance of this impact by other means than the CRA [28,29] as it appears also from our interviews. Several studies show that spousal caregivers report a greater burden on their schedule [30,31]. This can be related to family commitments such as childcare and social relationships exacerbating perceived burden and scheduling conflicts. Our caregiver population is predominantly spouses; this may explain the fact that this was the primary burden reported in our study. Financial and health impacts are noted in second and third position by caregivers, again following the same trend as the study cited above [27].

Based on SCNS-PC responses, their first need for help was for health care and information (score = 2.0). This is consistent with the literature, both in France (1.69) [32] and Germany (1.91) [33]. This corroborates what was declared by caregivers during the interviews: they need to be informed about everything concerning the patient and the disease. Lack of information is known to be a major source of stress for caregivers [34,35]. In our population, the proportion of caregivers with at least one USCN (93.7%) was higher than in the literature, where ranging from 16 to 68% [34,36] can be found. Our results are closer to the ones of Sklenarova et al, in which this score reached 85.6% [33]. The latter revealed that few variables were associated with caregivers' cancer-related USCNs, but that the number of patients' USCNs consistently predicted those of caregivers. Patient' USCN were not assessed in our study, which precludes the measurement of this correlation.

The positive aspects of caregiving are reflected in the CRA self-esteem dimension (score = 3.8), even if this score is slightly lower than the one in another study (4.51) [31]. Feelings of fulfillment and usefulness highlighted from the interviews are in tune with what is to be found in previous literature [3,10].

**Table 5. Similarities and topics addressed between patients and caregivers.**

| Similarities in the topics addressed | | |
|---|---|---|
| **Themes** | **Occurrences** | **Illustrative quotes from patients and caregivers** |
| **Caregiver burden** | | |
| Moral impact | 6 | C: "There is a stress... when he is not well, well I am worried. And now I find it hard... I can't handle it alone, I panic"<br>P: "She always tells me "be careful, be careful" and when there is something, she panics quickly, she panics." |
| Physical and health impact | 6 | C: "Now I take a medicine when I can't sleep, to help me sleep... Because sometimes I don't sleep all night, in fact when he doesn't sleep, when he has pain, I can't sleep..."<br>P: "When I spend my nights not sleeping or trying to sleep and then screaming because all of a sudden it hurts... But she never wanted us to split up, to have separate bedrooms. It's a... It's a mistake because I know that on the one hand I'm interrupting her sleep and on the other hand I'm worrying her." |
| Impact on schedule | 4 | C: "Well, I have no more free time."<br>P: "Because, well, in the end, he never has time. It's true, he's constantly with the kids." |
| Impact on studies/work | 3 | C: "I put my studies on hold for this year at least."<br>P: "and then she abandoned her studies." |
| The caregiver forgetting himself | 2 | C: "I almost didn't have the operation because I thought... it was an operation that was planned but as his condition had worsened in the meantime, I thought maybe I would cancel it. But in fact, nobody wanted me to cancel it, not my husband, not the doctors"<br>P: "I had to fight a little, because she didn't want to have her second prosthesis done right away: "well yes, but look at how you are" etc. I tell her you're right, the longer you wait, the greater the probability that I'll be worse off and the more complicated it will be to have surgery. I told her "you're right, the longer you wait, the greater the probability that I'll be worse off and the more complicated it will be to have the operation"." |
| Financial impact | 1 | C: "Financially yes, because [the patient] was supposed to go back to work at the end of the year. But she didn't, so now, inevitably, it puts a strain on the budget, which is quite limited."<br>P: "First we have to settle this financial matter... Because we can't, we can't pay everything... Well, no, no, after a while, it's expensive." |
| **The caregiver-patient relationship** | | |
| Unchanged relationship | 3 | C: "So our relationship is... Well, a couple that has spent many years together, with joyful moments and hard moments... but there is nothing really changed if you want on that side."<br>P: " Our relationship, it did not evolve so much, because since the time it does not evolve anymore." |
| Relationship improved | 3 | C: "Yeah, we actually talk more.... It's true that we may be closer than we were at times."<br>P: " I have the impression that our relationship has evolved, let's say, in the right direction." |
| **The caregiver's involvement** | | |

(*Continued*)

**Table 5.** (Continued)

| Themes | Occurrences | Illustrative quotes from patients and caregivers |
|---|---|---|
| **Similarities in the topics addressed** | | |
| The caregiver's presence | 4 | C: "Well, yes, I prefer to be there when she is there, yes, to be there in case she needs it. She's cyclothymic, so there can be times when, even if she's being treated, she takes her medication, but she can have moments when she's not feeling well. so that's why. . . I think I have my uses with her. Otherwise, I. . . I wouldn't have all these "worries". That's the way I look at it, I'd rather be there and not have any problems like we had before."<br>P: "He's concerned about seeing me sick, it's true that he doesn't really know, he's. . . he doesn't know, he doesn't know, he's even more concerned about leaving me alone while I'm having chemo because it's true that it didn't go very well the week of the chemo. So leaving me alone at home without any presence, he is concerned. It's true that unfortunately, I'm someone who is depressed, so it's true that I can have moments when I'm completely down. And then others where I am much more, how should I say, positive and full of energy but it's true that I. . . that, that anguishes him. . . I understand him eh, I understand him. But he is not serene, he is not serene at all." |
| Need of information of the caregiver | 3 | C: "The consultations, I systematically accompany him, because. . . I am curious, curious to know how it goes, the diagnoses etc."<br>P: "She's the one who follows up on the biological results, she's the one who takes them out, she reads them, she reads all my reports, and I don't read them. On the other hand, she reads everything. She goes to see, she wonders, she asks herself questions, she goes to see on the Internet etc." |
| **The positive aspects** | | |
| The caregiver is doing well | 2 | C: "I am in good health. Well, I also have my own pathology, so I take care of my health at the same time. . . But I'm fine."<br>P: "She seems to be doing well. I think she's doing well, yes." |
| New vision and philosophy of life | 1 | C:" In fact, on the contrary, this pathology allowed us to refocus on the essential. . . and to decelerate on our life in fact. . . To focus on the essential, to systematically ask ourselves the question and to say "What am I doing, what is it for? Do I need it?""<br>P: "We enjoy the present moments, in fact. We enjoy together and we don't restrict ourselves. It's a new vision of life." |
| **The future** | | |
| Facing uncertainty | 4 | C: "I wonder. . . I wonder how it will end. . . But I don't want to think about it, so probably wrong but I don't want to think about it. . . I'm afraid, I can't admit to being alone. . . It scares me, we've always been together. So I know that we are not eternal but I push away that idea. . . I push away the idea of death. He says to me: "you can do what you want", but I don't know, maybe I'll go to an EHPAD. . . I don't know if I'd like that. But I wouldn't want to stay in this house, alone. I don't see myself staying by myself."<br>P: "Lately things have been getting a little rushed, we had, we always kept a little hope, even knowing that it was a false hope. And so, we were kind of getting rid of that kind of, of problem. But yes, yes, she won't be able to stay alone in the house. There's 1700 square meters in total of space, even if we get a lot of things done, she won't be able to continue to do that alone. And will she feel like it afterwards, when I'm gone? Will she go through with what she says? Going to an EHPAD? because she says it thinking at the same time a little backwards, because she saw her mother in the same circumstances." |
| **Differences in the topics addressed** | | |
| **The caregiver-patient relationship** | | |

*(Continued)*

**Table 5.** (Continued)

| Similarities in the topics addressed | | |
|---|---|---|
| **Themes** | **Occurrences** | **Illustrative quotes from patients and caregivers** |
| Difference of feeling | 2 | C: *"We have moved away from each other."*<br>P: *"It's true that I have the impression that we are stronger... Well, it makes us more... more solid than before in fact."* |
| Hiding things from each other | 2 | P: *"I didn't tell [the caregiver] the diagnosis... the last operation, they found metastases. I said "no, that's too much", I only told my two older children, [the caregiver] I want to keep [him] from this "*<br>C → *the caregiver doesn't know that* |
| **The caregiver's need of help** | | |
| Different ways of feeling the caregiver's need for help | 2 | C: *"I don't need help. And as long as I don't ask, well, they don't come (her in-laws). But since I can manage everything myself, well no... but they ask anyway, they are there to ask if there is anything to do."*<br>P: *"Outside the house, many people have offered him: "Well, tell us, we'll come and help you, you tell us when you're doing that", but he doesn't call back. Because, "no, it's okay, I don't want to disturb you, I'll do my thing". And he has to admit, that he is, that he is... that he needs, he needs help."* |
| **Facing uncertainty** | | |
| Different ways of projecting | 1 | C: *"Projecting yourself is complicated, yes. We project ourselves completely differently because we are unable to project ourselves to a month. We don't know. Today, we live from day to day with the pathology."*<br>P: *"we look ahead, we have plans. We have travel plans, we have vacation plans, we have a lot of plans, professional reconversion plans for each other, so we move forward together, and here we are, we plan far ahead in fact and not in the short term."* |
| **Caregiver burden** | | |
| Impact on schedule | 1 | C: *"We don't dare... I know I didn't dare leave him too long... I cut off all my activities..."*<br>P: *"No, she didn't stop herself from doing certain things but she probably would have preferred that we had an easier life..."* |

The comparison of themes between the interviews with patients and caregivers and the analysis of the pairs individually reveals a large number of similarities. This shows that the patients have a fairly good representation of what the caregiver is going through, both in terms of the disruption of their habits and the different burdens they may feel, even though caregivers do not express their needs and difficulties. Hiding one's difficulties to appear strong to the patient is a known phenomenon [3,37]. However, this reluctance of caregivers to express their feelings has previously been shown to harm them [38]. This may explain why, as reported in various studies, a significant proportion of patients may underestimate the burden and difficulties of their caregivers [10,39]. Discrepancies in opinion were also found in an American study, with patients underestimating the difficulty of the psychosocial aspects of caregiving [40]. This is an important issue since this underestimation may significantly be related to lower quality of life and higher levels of depression and anxiety of the caregiver [39], and as it is known to be an important determinant of caregiver well-being [41]. Because patients and caregivers can influence each other [42], it is important to get the patient's opinion on what his caregiver is experiencing, and to promote communication between them. This would make it possible to highlight the differences in viewpoints between them, and improve their respective qualities of life. The main discrepancy between patients and caregivers concerns the evolution of their relationship dynamics. In the themes, caregivers report a deterioration in their

relationship while patients feel closer, and at the level of a pair, several couples report an evolution of their relationship in opposite directions. One study may partly explain this finding by suggesting that female caregivers, our predominant population, are more likely to describe themselves as grieving the relationship they previously shared with the patient [3]. Moreover, the fact that 80% of the caregivers interviewed were women and 60% of the patients were men could partly explain the differences in their points of view. According to Manne et al., a disparity between the views of patients and caregivers regarding their relationship is frequent [41]. In the literature, the results are also conflicting with studies reporting a deterioration [43] or an improvement in the relationship [3].

All this underlines the importance of supporting caregivers and identifying their psychosocial needs to be able to propose adjustments and direct them to the appropriate actors and structures, especially when the end of life approaches. In France, specific solutions exist (the leave for caregivers, the right to rest, etc.), but they are often ignored by caregivers [44]. Studies that have explored couple-based interventions demonstrated their benefit for both patient and caregiver [45,46]. For future research, it would be interesting to also investigate in more detail interventions based on parent/child and sibling relationships.

The principal strength of our study is its mixed-method approach. Some limitations were identified. Due to our limited sample size, additional data are needed for these results to be generalized. It would be of interest to submit the SCNS-PC and CRA questionnaires to the patients the same way we did with their caregiver, but it would mean having a psychometric validation of the questionnaires from their point of view. This may be an insightful work in the future. In addition, collecting data at different points in time would allow for an assessment of the evolution of experiences throughout the relationship. Indeed, it is known that the burden can evolve depending on the history of the disease [28].

Our study contributes to the growing literature in this area and shows that caregivers need as much support as patients. Our results are similar in many regards to other studies studying caregivers of patients with different types of cancer. This highlights the fact that caregivers' concerns are the same for many, despite populations with different characteristics and studies spread over time. Their consideration is not yet optimal. This study demonstrates the importance of considering both perspectives to better understand the caregiver's experience. Finally, it provides direction for implementing psychosocial interventions for the patient-caregiver dyad rather than interventions for caregivers or patients alone.

## Supporting information

**S1 File. Interview guides.**
(DOCX)

**S2 File. Additional quotes.**
(DOCX)

**S3 File. Raw quantitative anonymized data.**
(XLSX)

## Acknowledgments

The authors would like to warmly thank the participants of this study, as well as the medical teams of the medical oncology department of the CHU of Nantes for their help. We declare that this manuscript was posted as a preprint to https://www.researchsquare.com/article/rs-2037470/v2 (https://doi.org/10.21203/rs.3.rs-2037470/v2).

## Author Contributions

**Conceptualization:** Charlotte Grivel, Solange Pécout, Aurélie Lepeintre, Yann Touchefeu, Sonia Prot-Labarthe, Adrien Evin, Jean-François Huon.

**Data curation:** Charlotte Grivel, Pierre Nizet, Manon Martin.

**Formal analysis:** Manon Martin, Jean-François Huon.

**Investigation:** Adrien Evin, Jean-François Huon.

**Project administration:** Charlotte Grivel, Yann Touchefeu, Adrien Evin.

**Supervision:** Yann Touchefeu, Sonia Prot-Labarthe, Adrien Evin, Jean-François Huon.

**Validation:** Pierre Nizet, Solange Pécout, Aurélie Lepeintre, Sonia Prot-Labarthe, Adrien Evin, Jean-François Huon.

**Writing – original draft:** Charlotte Grivel.

**Writing – review & editing:** Pierre Nizet, Solange Pécout, Aurélie Lepeintre, Yann Touchefeu, Sonia Prot-Labarthe, Adrien Evin, Jean-François Huon.

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
