## [Decision Letter · Decision Letter 0]

2 May 2023

PONE-D-23-00504The experience of being a caregiver of patients with digestive cancer, from patients and caregivers’ perception: a mixed studyPLOS ONE

Dear Dr. HUON,

Thank you for submitting your manuscript to PLOS ONE. After careful consideration, we feel that it has merit but does not fully meet PLOS ONE’s publication criteria as it currently stands. Therefore, we invite you to submit a revised version of the manuscript that addresses the points raised during the review process.

We look forward to receiving your revised manuscript.

Kind regards,

Sidrah Nausheen, FCPS

Academic Editor

PLOS ONE

Reviewers' comments:

Reviewer's Responses to Questions

**Comments to the Author**

1. Is the manuscript technically sound, and do the data support the conclusions?

Reviewer #1: Yes

Reviewer #2: Yes

2. Has the statistical analysis been performed appropriately and rigorously? 

Reviewer #1: Yes

Reviewer #2: Yes

3. Have the authors made all data underlying the findings in their manuscript fully available?

Reviewer #1: Yes

Reviewer #2: Yes

4. Is the manuscript presented in an intelligible fashion and written in standard English?

Reviewer #1: Yes

Reviewer #2: Yes

5. Review Comments to the Author

Reviewer #1: Mixed method approach to the study provides detailed insight to the caregivers and patients perspectives. With the growing incidence of cancers all over the world, this is important aspects to look at.

The author has explained it very well.

Reviewer #2: It is a nice and rare topic to address the problems faced by caregivers especially to the cancer patients who might be terminally ill.

The methodology is up to the mark.

Number of participants are not impressive.

Was there any method adopted to address the problems faced by caregivers especially regarding the sleep deprivation and dealing with the stress.

6. PLOS authors have the option to publish the peer review history of their article (what does this mean?). If published, this will include your full peer review and any attached files.

Reviewer #1: No

Reviewer #2: **Yes: **Syed Muhammad Ali

---

## [Author Response · Author response to Decision Letter 0]

15 May 2023

PONE-D-23-00504

The experience of being a caregiver of patients with digestive cancer, from patients and caregivers’ perception: a mixed study

PLOS ONE

Response to Editors and Reviewers

Editors

A1. Please ensure that your manuscript meets PLOS ONE's style requirements, including those for file naming. 

Q1: We thank the editor for this remark. Changes were made accordingly (level 2 and 3 headings’ font, affiliation formatting…).

A2. Thank you for stating the following financial disclosure. Please clarify the sources of funding (financial or material support) for your study. If you did not receive any funding for this study, please state: “The authors received no specific funding for this work.”

Q2: Thank you for this relevant remark. No specific funding was received for this work so the disclosure was modified as requested in the cover letter.

A3. Please include your full ethics statement in the ‘Methods’ section of your manuscript file. In your statement, please include the full name of the IRB or ethics committee who approved or waived your study, as well as whether or not you obtained informed written or verbal consent. If consent was waived for your study, please include this information in your statement as well.

Q3: As requested, an Ethics statement was added to the Methods section, including the full name of the committee and how consent was obtained.

A4. In your Data Availability statement, you have not specified where the minimal data set underlying the results described in your manuscript can be found. PLOS defines a study's minimal data set as the underlying data used to reach the conclusions drawn in the manuscript and any additional data required to replicate the reported study findings in their entirety. All PLOS journals require that the minimal data set be made fully available. For more information about our data policy, please see http://journals.plos.org/plosone/s/data-availability.

Q4: Thank you for pointing that out.

For quantitative data, raw data were added as Supplemental material (S3).

For data collected as part of the qualitative research, we made excerpts of the transcripts relevant to the study available in the manuscript and in the supplemental data. Sharing the full transcripts would violate the agreement to which the participants consented, as explained in the Data Availability Statement.

A5. Please review your reference list to ensure that it is complete and correct. If you have cited papers that have been retracted, please include the rationale for doing so in the manuscript text, or remove these references and replace them with relevant current references. 

Q5: After verification of the references, it seems that the list is correct and that no cited paper was retracted.

Reviewers

A6: Reviewer #1: Mixed method approach to the study provides detailed insight to the caregivers and patients perspectives. With the growing incidence of cancers all over the world, this is important aspects to look at. The author has explained it very well.

Q6: We thank the Reviewer 1 for this thoughtful comment and their interest in our work.

A7: Reviewer #2: It is a nice and rare topic to address the problems faced by caregivers especially to the cancer patients who might be terminally ill. The methodology is up to the mark. 

Q7: We thank the Reviewer 2 for this thoughtful comment and their interest in our work.

A8: Number of participants are not impressive.

Q8: We agree with Reviewer 2, as already highlighted in the study limitations.

A9: Was there any method adopted to address the problems faced by caregivers especially regarding the sleep deprivation and dealing with the stress.

Q9: Having re-examined the transcripts of the caregivers who reported this type of problem, it appears that the methods sought to remedy it were not frequently discussed during the interviews. However, three caregivers (C15, C26, C53) stated they were taking some pills, having naps during the day, or watching TV during the night when having trouble sleeping. One of them preferred sleeping in a room alone to have better sleep. This was precised in the manuscript (page 17).

Figure files were uploaded to the Preflight Analysis and Conversion Engine (PACE) digital diagnostic tool.

We hope that the revised manuscript is now suitable for publication and we would like to thank the Reviewers and Editors.

---

## [Editor Report · Decision Letter 1]

5 Jun 2023

The experience of being a caregiver of patients with digestive cancer, from patients and caregivers’ perception: a mixed study

PONE-D-23-00504R1

Dear Dr.Jean-François HUON,

We’re pleased to inform you that your manuscript has been judged scientifically suitable for publication and will be formally accepted for publication once it meets all outstanding technical requirements.

Kind regards,

Sidrah Nausheen, FCPS

Academic Editor

PLOS ONE
---

## [Editor Report · Acceptance letter]

13 Jul 2023

PONE-D-23-00504R1 

The experience of being a caregiver of patients with digestive cancer, from patients and caregivers’ perception: a mixed study 

Dear Dr. Huon:

I'm pleased to inform you that your manuscript has been deemed suitable for publication in PLOS ONE. Congratulations! Your manuscript is now with our production department. 

Kind regards, 

on behalf of

Dr. Sidrah Nausheen 

Academic Editor

PLOS ONE